# Efficient Fourier Neural Operators by Group Convolution and Channel Shuffling

**Myungjoon Kim,**[*] **Junhyung Park,**[*] **Jonghwa Shin**
Department of Materials Science and Engineering
KAIST
Daejeon, Republic of Korea
`{vcxzx, ekdldkqkr, qubit}`@kaist.ac.kr

## Abstract

Fourier neural operators (FNOs) have emerged as data-driven alternatives to conventional numerical simulators for solving partial differential equations (PDEs). However, these models typically require a substantial number of learnable parameters. In this study, we explore parameter-efficient FNO architectures through modifications in their width, depth, and applications of group convolutiopn and channel shuffling. We perform benchmark on different problems on learning the operator of Maxwell's equations and Darcy flow equations. Our approach leads to significant improvement in prediction accuracy for both small and large FNO models. The proposed methods are widely adaptable across various problem types and neural operator architectures, aiming to boost prediction accuracy.

## 1 Introduction

Science and engineering problems are often characterized by partial differential equations (PDEs). To solve these complex PDEs, numerical simulations, such as the finite difference method (FDM) or the finite element method (FEM), are employed. These traditional approaches solve PDEs through particular algorithms that discretize space and time, often requiring substantial computational resources. Such a computationally demanding part is a bottleneck, hindering the rapid calculation and analysis of the system.

Recently, the advancement of deep neural networks has made data-driven models a promising approach for solving PDEs. While convolutional neural networks (CNNs) have been applied to these challenges, they often fall short for PDE problems due to their inherent design for mapping between Euclidean spaces. A new class of neural PDE solvers, known as *neural operators*, has emerged to address these limitations. Among various methods, Fourier neural operators (FNOs) stand out as a solution for solving complex PDEs, offering high accuracy and rapid prediction capabilities. Numerous research aimed at exploring diverse applications and enhancing functionalities of FNOs.

In this work, we implement an architectural modification to FNOs aimed at increasing both efficiency and accuracy. Through modifications in width, depth, and applying group convolution and channel shuffling, inspired by the traditional CNN architectures, we can find the optimal architecture with best performance. The main contribution of our works are summarized as follows:

- We present an adjustment to the architecture of the Fourier layer, modifying its width and depth, incorporate group convolution along with channel shuffling to improve performance, and show our approach is memory and computation efficient way to improve large FNO models.

- Thorough benchmarking and detailed analysis show that modifications to the architecture significantly boost model performance. The benchmark results highlight the importance of selecting the appropriate model for accurately predicting the solution field across different PDE problems.

---

[*]equal contribution

## 2 RELATED WORKS

**Parameter efficient architectures in convolutional neural networks**   Since the introduction of the *ResNet* architecture (He et al., 2016), numerous modifications have been proposed to enhance its representational capacity while maintaining the complexity of neural networks. Zagoruyko & Komodakis (2016) suggested *WideResNet* and explored how increasing the width of the network could contribute to higher accuracy in image classification tasks. The *ResNeXt* architecture employs group convolution to enhance the *cardinality* of its layers, contributing to its effectiveness and efficiency (Xie et al., 2017). *ShuffleNet* extends the application of group convolution to include pointwise operations and incorporates channel shuffling to optimize performance and efficiency (Zhang et al., 2018).

**Fourier neural operators**   FNOs are advanced computational models that efficiently solve PDEs by leveraging Fast Fourier transforms to parameterize and predict complex dynamics. FNOs learn mapping between infinite-dimensional function spaces (Li et al., 2020), thus they are well-suited as surrogate models for conventional PDE simulators. Recently, numerous variants of the FNO have been proposed to enhance data and memory efficiency (Rahman et al., 2022; Li et al., 2022; 2021; Helwig et al., 2023). Notably, Factorized Fourier Neural Operator (F-FNO) introduces a dimension-wise factorization approach (Tran et al., 2022). This was achieved by representing 2D Fourier domains through the summation of two orthogonal 1D Fourier transforms.

**Surrogate models for numerical simulations**   Numerous neural network-based methods have been developed to solve PDEs (Kovachki et al., 2023). In the field of optics, researchers have tackled the problem of field distribution prediction using CNNs (Wiecha & Muskens, 2019; Chen et al., 2022; Lim & Psaltis, 2022). More recently, specialized variants of FNOs designed to solve Maxwell's equations have shown superior performance over traditional CNNs (Gu et al., 2022; Augenstein et al., 2023).

## 3 METHODS

### 3.1 PARAMETER-EFFICIENT FOURIER NEURAL OPERATORS

The Fourier integral operator in the original Fourier neural operator (Li et al., 2020) is defined as

$$\mathcal{L}^{(l)}\left(v^{(l)}\right) = \sigma\left(W^{(l)} z^{(l)} + \mathcal{K}^{(l)}(z^{(l)})\right) \tag{1}$$

where $\mathcal{K}^{(l)}$ represents the learnable kernel integral operator using the Fourier transform $\mathcal{F}$ and inverse Fourier transform $\mathcal{F}^{-1}$, defined as

$$\mathcal{K}^{(l)} = \mathcal{F}^{-1}(R^{(l)} \cdot \mathcal{F}) \tag{2}$$

The parameter numbers in FNO is $O\left(LH^2 M^D\right)$, where $L$, $H$, and $M$ represents the number of Fourier layer, the channel, the Fourier modes being kept, respectively. $D$ denotes the dimension of the problem.

In our proposal, we apply modifications to FNO models to enhance its parameter efficiency. Initially, we tune the depth (the number of Fourier layers) and the width (the number of channels) of the model. Specifically, we incorporate pointwise group convolution and channel shuffling into the architecture. The design of our modified version of the architecture is depicted in Figure 1. Within each Fourier layer, we implement group convolution in both the frequency and spatial domains. This technique reduces the number of parameters to $O\left(\frac{LH^2 M^D}{g}\right)$, where $g$ represents the group number. By doing so, we decrease the model's complexity by a factor of $g$, enhancing its parameter efficiency. Moreover, we utilize channel shuffle operations at every Fourier layer to boost the representation capability by enabling effective crosstalk among features. These operations involve reshaping the feature dimension into $(g, m)$, transposing it, and then flattening it back to the input format for the subsequent layer.

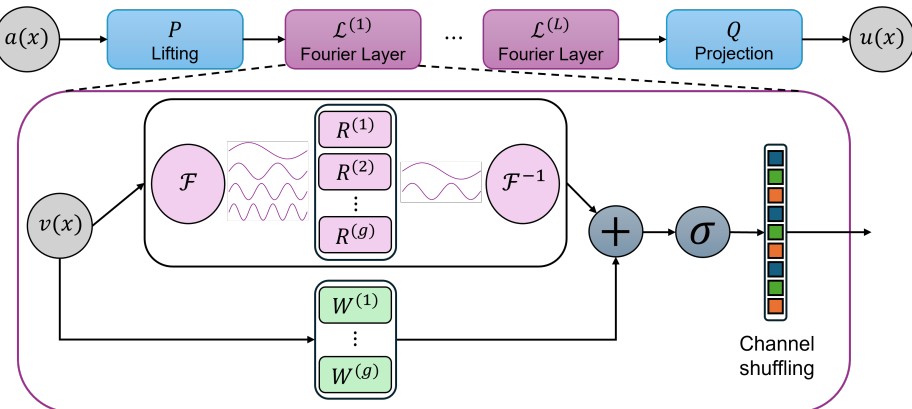

Figure 1: The architecture of the efficient Fourier neural operator. $\mathcal{F}$ and $\mathcal{F}^{-1}$ represents fast Fourier transform and inverse fast Fourier transform, respectively. The linear transformations both in spatial and frequency domain are done using pointwise group convolution. The channels are shuffled after the nonlinear activation.

## 4 EXPERIMENT

### 4.1 DATASET

**Periodic metagrating**  In nanophotonics, periodic nanostructures with the size comparable to wavelength can be used for diffraction control devices. These have been implemented to steer light beams (Sell et al., 2017) in specific directions, and phase masks for photolithographic process (Nam et al., 2022). The optical responses are governed by Maxwell's equations.

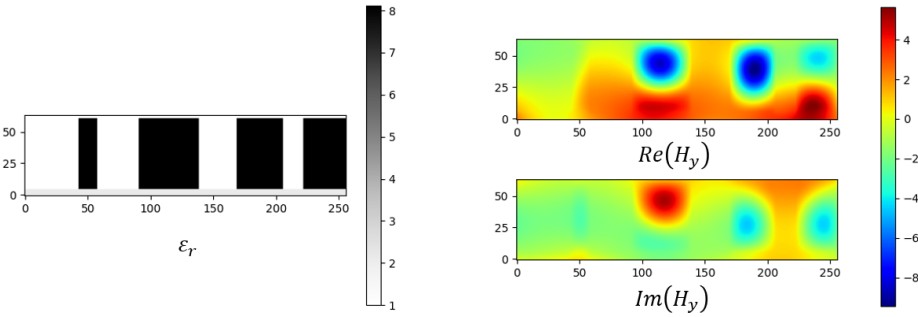

Figure 2: A metagrating structure and corresponding magnetic field distribution.

We use a metasurface dataset originally provided by Chen et al. (2022), available in the online database (Jiang et al., 2020). We sampled 12,000 and 1,500 data for training and validation, respectively. Figure 2 illustrates an example of the metagrating structure along with the fields computed by the FDM solver. The system comprises periodic boundaries along the x direction and perfectly matched layers, which are artificial layers designed to absorb all radiated fields, along the y direction.

These structures are converted to $[1, 64, 256]$ image format, where each pixel corresponds to a mesh size of $6.25nm \times 6.25nm$. The incident light has a wavelength of 1050 nm and is polarized in the transverse magnetic (TM) mode. The relative permittivity of the materials ranges from 6.12 to 11.90, and there is a substrate with a permittivity of 2.10.

**Darcy Flow**  We demonstrate an additional examples using parameter-efficient FNO to solve different type of PDE. Darcy flow equations describe the movement of fluids through porous media, relating fluid velocity to the pressure gradient, permeability of the medium, and fluid viscosity. We

use standard two-dimensional Darcy Flow equation on the unit box. The dataset consists of 1000 training samples and 200 validation samples, all arranged in a uniform grid of $241 \times 241$.

## 4.2 RESULTS

**Training**   We use F-FNO for all experiments. For training, we use AdamW optimizer (Loshchilov & Hutter, 2017) with an initial learning rate of 0.001 and a batch size of 32. For all models, we use exponential learning rate scheduling and the total number of epochs is set to 200. GeLU serves as the activation function, and we maintain $[12, 12]$ Fourier modes for all experiments. For the loss function and the prediction accuracy metric, we use the normalized mean absolute error (NMAE) defined as:

$$NMAE = \frac{1}{N} \sum_{i=1}^{N} \frac{\|\hat{\mathbf{y}} - \mathbf{y}\|_1}{\|\mathbf{y}\|_1} \tag{3}$$

where $\mathbf{y}$ and $\hat{\mathbf{y}}$ are the predicted field values and ground truth, respectively, and $\|\mathbf{y}\|_1$ is the $L_1$ norm of the field distribution.

**Depth vs. Width**   In our initial experiment, we evaluated the impact of depth and width while maintaining a fixed number of parameter of small (0.56M) and large (2.2M) models. We tested configurations with 5, 10, 15, and 20 Fourier layers, adjusting the number of channels for each layer accordingly. The results showed that a setup with 15 layers and 27 channels for the small model, and 10 layers with 64 channels for the large model, exhibits the best performance. From these findings, we conclude that there are specific optimal values for the number of layers and channels that produce the highest efficiency within a given parameter budget.

Table 1: The effect of depth and width on F-FNO and its variants. The total parameter numbers are maintained at 0.56M and 2.2M.

| Model | Layers | Channels | Params. | NMAE |
|-------|--------|----------|---------|------|
| F-FNO | 5 | 46 | 0.56M | 0.054 |
| F-FNO | 10 | 32 | 0.56M | **0.036** |
| F-FNO | 15 | 27 | 0.56M | 0.041 |
| F-FNO | 20 | 23 | 0.56M | 0.044 |
| F-FNO | 5 | 92 | 2.2M | 0.041 |
| F-FNO | 10 | 64 | 2.2M | **0.028** |
| F-FNO | 15 | 54 | 2.2M | 0.035 |
| F-FNO | 20 | 46 | 2.2M | 0.038 |

**Group convolution and channel shuffling** We explore the impact of group convolution and channel shuffling on model performance. As in the earlier experiment, we performed benchmarks with both small and large models. We kept the number of parameters constant, increasing the number of groups and channels simultaneously (Table 2). In both cases, the models achieved their best performance when the group number is set to 8. As the number of groups increases, it enhances the feature representation capability, which in turn improves the overall performance of the model.

Table 2: Application of group convolution in F-FNO models, where $g$ denotes the number of groups. Each model consists of 10 Fourier layers.

| Model | Channels | Param. | NMAE |
|-------|----------|--------|------|
| F-FNO | 32 | 0.56M | 0.036 |
| F-FNO ($g = 2$) | 44 | 0.56M | 0.033 |
| F-FNO ($g = 4$) | 56 | 0.56M | 0.032 |
| F-FNO ($g = 8$) | 72 | 0.56M | **0.030** |
| F-FNO | 64 | 2.2M | 0.028 |
| F-FNO ($g = 2$) | 88 | 2.2M | 0.025 |
| F-FNO ($g = 4$) | 116 | 2.2M | 0.023 |
| F-FNO ($g = 8$) | 144 | 2.2M | **0.022** |

Overall, it is evident that neural operators perform best when their width, depth, and group numbers are set optimally within the given parameter budget constraints. Prediction accuracy increases by 33% and 31% through the appropriate selection of width/depth ratios, and further improvements of 16% and 21% are achieved by incorporating group convolution and channel shuffling in small (0.56M) and large (2.2M) models, respectively. These results highlight the importance of the selection of architectures to maximize model performance.

We also evaluated the impact of group convolution on FNO models while keeping the number of channels (Table 3). In small FNO models, we observed a performance decline as the number of parameters was reduced by increasing the number of groups. On the other hand, in large models, performance enhanced with the addition of group convolutions These suggest that for large models,

sparse connection may enhance generalization capability. We anticipate that by integrating memory-efficient models (Rahman et al., 2022), we can develop more efficient FNO models.

Table 3: Impact of pointwise group convolution on small and large F-FNO models with identical channel numbers. Each model has 10 Fourier layers.

| Model | Channels | groups | Param. | NMAE |
|-------|----------|--------|--------|------|
| F-FNO | 32 | 1 | 0.55M | 0.036 |
|       |    | 2 | 0.30M | 0.044 |
|       |    | 4 | 0.17M | 0.045 |
|       |    | 8 | 0.11M | 0.055 |
| F-FNO | 144 | 1 | 11M | 0.027 |
|       |     | 2 | 5.9M | 0.024 |
|       |     | 4 | 3.4M | 0.023 |
|       |     | 8 | 2.1M | 0.022 |

**Darcy flow**   The effects of group convolutions and channel shuffling on the Darcy flow dataset are reported in Table 4. The prediction accuracy improves with an increase in the number of groups. Similar trends suggest that our method could be broadly applicable to various differential equations.

Table 4: Impact of pointwise group convolution on Darcy Flow equation

| Model | Channels | groups | Param. | NMAE |
|-------|----------|--------|--------|------|
| F-FNO | 32 | 1 | 0.55M | 0.083 |
|       | 44 | 2 | 0.56M | 0.036 |
|       | 56 | 4 | 0.52M | **0.029** |

## 5   CONSLUSION

In this study, we introduce parameter-efficient FNOs, modified neural operator achitectures capable of solving PDE problems. Leveraging previous development in conventional CNNs, we investigated the effectiveness of changing width, depth, and group convolutions and channel shuffling in FNOs. Our proposed methods achieve superior representational capabilities by increasing the number of channels while maintaining the overall parameter count. We conducted two benchmark problems, Maxwell's equations and Darcy flow equation. The architectural modifications significantly improve prediction accuracy. Moving forward, we aim to tackle a wider array of PDE systems, including diffusion processes, and Navier-Stokes equations.

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

## A  ADDITIONAL EXPERIMENTS

Table A1: The effect of pointwise group convolution on FNO models with same channels. All models have 10 Fourier layers.

| Model | Channels | groups | Param. | NMAE |
|-------|----------|--------|--------|------|
| FNO | 32 | 1 | 6.0M | 0.048 |
|  |  | 2 | 3.0M | 0.050 |
|  |  | 4 | 1.5M | 0.055 |
|  |  | 8 | 0.78M | 0.071 |
| FNO | 72 | 1 | 30M | 0.038 |
|  |  | 2 | 15M | 0.038 |
|  |  | 4 | 7.7M | 0.036 |
|  |  | 8 | 4.0M | 0.039 |
| FNO | 144 | 1 | 120M | 0.031 |
|  |  | 2 | 5.9M | 0.031 |
|  |  | 4 | 0.14M | 0.025 |
|  |  | 8 | 0.07M | 0.024 |

**Generalization with FNO**  We examined the effect of pointwise group convolution and channel shuffling on the original FNO architectures. Within the constraints of parameter number budgets or with the same number of channels for large models, pointwise group convolution and channel shuffling enhance FNO models. Therefore, our approach appears to consistently boost the performance of various FNO models.

