# OpenReview forum: "Efficient Fourier Neural Operators by Group Convolution and Channel Shuffling"
_ICLR.cc/2024/Workshop/AI4DiffEqtnsInSci — AI4DiffEqtnsInSci @ ICLR 2024 Poster_

### Official Review · Reviewer_q5Xb · 2024-02-24
**Efficient fourier neural operators by group convolution and channel shuffling**

**Rating:** 7
**Confidence:** 2

**Review:**

## General comments
The paper introduces enhancements to Fourier Neural Operators, incorporating
group convolution and channel shuffling, aiming for improved efficiency and
accuracy in solving partial differential equations. The approach is evaluated
through benchmarks, showing significant performance enhancements. The
methodology combines theoretical foundations with empirical validation. The
document is structured to clearly outline the contributions, facilitating
understanding among readers with interests in scientific computing and data-
driven modeling.
The work is ready for publication, offering valuable insights and advancements
without the need for further modifications.

---

### Official Review · Reviewer_Acbj · 2024-02-24
**This paper presents a development of architecture in Factorized - Fourier Neural Operator (F-FNO) to reduce the computational overheads and so prediction performance in a case study. It introduces group convolution to the F-FNO framework to decrease computational complexity by a factor of the number of groups. The authors test the performance evaluation for Maxwell PDEs as a preliminary benchmark investigation.**

**Rating:** 6
**Confidence:** 5

**Review:**

The study conducted benchmarking to demonstrate progress in both computational complexity and prediction accuracy by combining group convolution with hyperparameter optimizations, which is a strength of this paper. However, the authors' claim of a reduction of N-MSE from 0.054 to 0.031 should be clarified. It appears that the benefit of reduction by introducing the group convolution should be calculated as the reduction from 0.043 to 0.031, resulting in a 27.9% reduction, especially concerning the same number of Fourier layers (i.e., block) used.

While the theoretical aspects are clear, the paper lacks clarity regarding the practical memory efficiency gained during actual computations. However, the dataset and problem are well-defined, aiding understanding.

The authors justify its originality in combining group convolution with the F-FNO architecture. This approach contributes to the field by proposing a novel method for addressing computational efficiency and prediction accuracy simultaneously.

The use of the proposed method can enhance prediction accuracy and computational efficiency, as demonstrated in the study. The performance improvements observed in a case study involving Maxwell's equations in the nanophotonic structure domain underscore the importance of accelerating FNO to other scientific domains governed by partial differential equations (PDEs). However, it remains unclear whether the proposed architecture can achieve similar improvements in prediction accuracy across other domains, which warrants further investigation.

---

### Meta-Review · Area_Chair_RwXN · 2024-03-01

**Recommendation:** Accept (Poster)

**Metareview:**

Thanks to reviewers for their feedback. The paper proposes the advantage of group convolution and channel shuffling in neural operator based models to improve model computational efficiency as well as accuracy. I encourage the authors to carefully review the comments of reviewers and clearly address them. Due to workshop limited page number, consider adding more details to appendices. I vote accept to this paper.

---

### Decision · Program_Chairs · 2024-03-02

Accept (Poster)